# Comparison of Classifier Calibration Schemes for Movement Intention Detection in Individuals with Cerebral Palsy for Inducing Plasticity with Brain–Computer Interfaces

**DOI:** 10.3390/s25237347

**Published:** 2025-12-02

**Authors:** Mads Jochumsen, Cecilie Sørenbye Sulkjær, Kirstine Schultz Dalgaard

**Affiliations:** Department of Health Science and Technology, Aalborg University, 9260 Gistrup, Denmark

**Keywords:** brain–computer interface, movement-related cortical potential, cerebral palsy, movement intention, neurorehabilitation, neural plasticity, EEG

## Abstract

Brain–computer interfaces (BCIs) have successfully been used for stroke rehabilitation by pairing movement intentions with, e.g., functional electrical stimulation. It has also been proposed that BCI training is beneficial for people with cerebral palsy (CP). To develop BCI training for CP patients, movement intentions must be detected from single-trial EEG. The study aim was to detect movement intentions in CP patients and able-bodied participants using different classification scenarios to show the technical feasibility of BCI training in CP patients. Five CP patients and fifteen able-bodied participants performed wrist extensions and ankle dorsiflexions while EEG was recorded. All but one participant repeated the experiment on 1–2 additional days. The EEG was divided into movement intention and idle epochs that were classified with a random forest classifier using temporal, spectral, and template matching features to estimate movement intention detection performance. When calibrating the classifier on data from the same day and participant, 75% and 85% classification accuracies were obtained for CP- and able-bodied participants, respectively. The performance dropped by 5–15 percentage points when training the classifier on data from other days and other participants. In conclusion, movement intentions can be detected from single-trial EEG, indicating the technical feasibility of using BCIs for motor training in people with CP.

## 1. Introduction

Brain–computer interfaces (BCIs) enable people with severe motor impairments to control external technology using only voluntarily produced brain activity [1]. It has been shown to enable control of speller devices, wheelchairs, robotic arms, and game applications, amongst other things [2]. Since BCIs only rely on brain activity, and the user’s ability to produce control signals that can be extracted from the brain activity, it has been used by people with various neurological disorders associated with severe motor impairments such as amyotrophic lateral sclerosis and spinal cord injury. Over the past years, BCIs have also been studied extensively within stroke rehabilitation [3,4], where BCIs have been shown to induce neural plasticity [5,6], which is the proposed underlying mechanism for motor learning [7]. BCI-based training has consistently been shown to improve motor function in stroke patients [3,4,7,8]. The rehabilitation after stroke for improving motor function is similar to the habilitation for promoting motor function in individuals with cerebral palsy (CP) [9]. Therefore, it is hypothesized that BCI-based training can induce plasticity in people with CP [10,11] and hence improve motor function in people with CP. It has also been shown that BCI-controlled hand exoskeleton training led to functional improvements in clinical scales for children and adolescents with CP [12], and in a case study it was reported that gait function improved after BCI-controlled electrical stimulation of the tibialis anterior muscle [13]. In BCI-based training, Hebbian-associated plasticity is induced by pairing movement-related brain activity with temporally coupled congruent somatosensory feedback from, e.g., electrical stimulation [14,15], exoskeletons, or rehabilitation robots [16,17]. The BCI records the ongoing electrical brain activity (electroencephalography—EEG) in which control signals need to be extracted. For inducing plasticity, two types of control signals associated with movement preparation can be extracted: movement-related cortical potentials (MRCPs) [18] and event-related desynchronization (ERD) [19]. Once the control signal is detected by the BCI, a trigger is immediately sent to the external device that can replicate the intended movement and hence elicit congruent somatosensory feedback from the movement which is temporally linked. To maximize the induction of neural plasticity, the somatosensory feedback should reach the cortex during the most negative phase of the MRCP [20], which is during the onset of the movement; hence, the movement intention should be detected with a very short latency to allow the afferent feedback to reach the cortex, which takes approximately 25 and 50 ms for the upper and lower limbs, respectively [21]. This is possible using both MRCPs and ERD which can be detected prior to the movement onset, with detection accuracies ranging between approximately ~70–90% [22,23,24,25,26,27]. MRCPs and ERD have been detected in able-bodied participants and in people with various neurological conditions causing motor impairments such as stroke [27,28,29], spinal cord injury [30,31], amyotrophic lateral sclerosis [32], multiple sclerosis [33,34,35], Parkinson’s Disease [36], and CP [37,38]. Most of the existing literature relating to movement intention detection in people with CP has focused on detecting ERD. In a recent feasibility study for using BCI-activated electrical stimulation as a training intervention, it was shown that children and adolescents with CP showed abilities to control a BCI with detection accuracies around 75% [11]. This is consistent with previous work that has shown that the majority of the included CP participants can modulate their ERD with a detection performance significantly higher than that of a chance level [38,39]. Moreover, it has been reported that BCI performance by CP users improves with training over multiple sessions [40,41], although factors such as concentration, impairment, and motivation affect the performance [11,42].

A single study has focused on MRCP detection from single-trial EEG in adolescents with CP where attempted; foot movements were detected with accuracies ranging between 70 and 80% when training and testing on data from the same day and same participant [37]. Also, a feature analysis was performed where it was reported that temporal features of the MRCP, or the combination of temporal and spectral features (ERD), were associated with higher classification accuracies than spectral features alone [37]. In the existing work within movement intention decoding in individuals with CP, calibration of the BCI has been performed immediately prior to it being used or when using the data from the same session to train and evaluate a classifier for estimating the BCI performance. Individualizing the BCI to the user immediately before it is used generally leads to the best BCI performance, but this approach requires a calibration process where training data needs to be collected before the detection algorithms are calibrated and the BCI is ready to be used, which can last around 15–30 min. For CP users with limited attention span, this can be an issue, and it reduces the likelihood of the technology being adopted in clinical practice if it takes too much time before the actual training regimen starts [43]. It is possible to overcome this issue by using a user-independent classifier generated from data from multiple patients and study participants where no training data from the individual user is needed [33,36], or when using previously recorded data from the user, if that is available [13,33,36,44,45]. In summary, there is a lack of studies that have investigated different BCI calibration schemes for detection of movement intentions in individuals with CP. Also, there is no evidence on detection of MRCPs associated with hand movements in individuals with CP. Therefore, the aim of this study was to validate existing findings in other studies and compare different calibration scenarios to investigate the effect of decoding movement intentions from single-trial EEG in individuals with CP and able-bodied participants with a focus on using it for inducing neural plasticity as a means for motor training in individuals with CP. This was investigated for the upper and lower extremities for individuals with CP. The BCI performance was estimated by classifying between movement intentions and idle brain activity in four scenarios in offline analyses: (1) training and testing the classifier with data from the same session and participant, (2) training and testing the classifier within the same participant but on data from separate sessions, (3) training the classifier on data from other participants and testing it on data from a separate participant, and (4) training the classifier on data from able-bodied participants and testing it on data from individuals with CP. Moreover, the importance of different feature types was estimated.

## 2. Materials and Methods

### 2.1. Participants

Five individuals with CP were recruited for the study (see Table 1 for an overview of the CP participants). The first four CP participants participated in two experimental sessions on two separate days while the fifth CP participant participated in one experimental session. Moreover, 15 able-bodied participants, 8 women and 7 men (mean age: 29 ± 9 years), participated in three experimental sessions on three separate days. All procedures were approved by the North Denmark Region Committee on Health Research Ethics (protocol numbers: N-20230015, N-20230025, and N-20240040), and all participants provided their written informed consent prior to participation.

### 2.2. Experimental Setup

The participants were seated in a comfortable chair, or their wheelchair, facing a computer screen positioned approximately one meter away. The participants performed ballistic movements with low force, to minimize muscle fatigue, while continuous EEG was recorded. The movements were visually cued by a computer, and when the cue to move was shown, a digital trigger was sent to the EEG amplifier for synchronizing the movements with the cues. The visual cues were shown 3 s in advance so that the participant knew when to initiate the movement. The CP participants performed wrist extensions and ankle dorsiflexions with their most affected side (right for all participants), and the able-bodied participants performed wrist extensions with their right hand. For further synchronization between movement onset and the EEG, EMG was recorded from the CP participants since it was expected that they could have difficulties timing the movement onset correctly with respect to the visual cue to move, especially if they started to fatigue (see Figure 1). No EMG was recorded from the able-bodied participants since it was assumed that they could initiate the movement consistently with respect to the visual cue. The participants performed a movement every 10 s. The able-bodied participants performed 50 wrist extensions using the right side, while the CP participants performed up to 75 wrist extensions and 75 ankle dorsiflexions using the right side. For the CP participants, the movements were performed in blocks of 25 repetitions of the same movement type followed by a break, and the order of blocks alternated between movements of the wrist and ankle, e.g., 25 repetitions of wrist extension followed by a break and 25 ankle dorsiflexions. The break between each block was between 2 and 5 min, depending on the participant’s preference. A 3 min recording of idle activity was placed after the first two blocks and after the first four blocks, i.e., two blocks of idle activity were recorded. An identical experiment was performed on a separate day for the CP participants and on two separate days for the able-bodied participants, i.e., data from two and three recording days were obtained from CP and able-bodied participants, respectively.

### 2.3. Recordings

Five channels of continuous EEG were recorded from C3, C1, Cz, C2, and C4 using active electrodes (g.GAMMAcap and g.SCARABEO electrodes, g.Tec, Schiedlberg, Austria). The channels were chosen since they covered the primary motor cortex. The five channels were referenced to the right earlobe and grounded at AFz. During the recordings, the participants were instructed to sit as still as possible and avoid blinking and activating facial muscles three seconds prior to the movement onset until one second after. For the CP participants, EMG was recorded over the extensor muscles in the right forearm and over the tibialis anterior muscle in the right leg. A bipolar derivation from each limb was recorded from two electrodes placed on the belly of the muscle and the ground electrodes were placed on tibia and ulna for the recordings from tibialis anterior and wrist extensor muscles, respectively. The EEG and EMG were sampled with 512 Hz using the same amplifier (g.HIAMP, g.Tec, Schiedlberg, Austria). Moreover, the digital trigger from the cueing program was also recorded using the same amplifier.

### 2.4. EMG Onset Detection and Pre-Processing

Initially, a 4th order zero-phase shift Butterworth bandpass filter was used to filter the EMG between 20 and 100 Hz and a Notch filter between 48 and 52 Hz was applied, after which the filtered EMG was rectified before starting the onset detection. The movement onsets were identified using an open-source toolbox (emgGO, [46]) using a double threshold algorithm. Each identified onset was visually inspected and if an onset was not correctly identified it was corrected. The EEG was bandpass filtered between 0.1 and 30 Hz using a 4th order zero-phase shift Butterworth bandpass filter. Based on the identified movement onsets, or digital triggers for the able-bodied participants, the continuous EEG was divided into two types of epochs: Movement intention epochs (−2 to 0 s with respect to the movement/cue onset) and 2 s idle activity epochs, which were were extracted from the 3 min recordings for the CP participants. For the able-bodied participant, the idle activity epochs were extracted from 3 to 5 s prior to the movement onset. EEG epochs with amplitudes exceeding 150 µV were rejected from further analysis. For the CP participants performing wrist movements, 3 ± 4 (mean ± standard deviation) and 1 ± 1 epochs were rejected from day 1 and 2, respectively. For the ankle movements, 9 and 7 ± 10 epochs were rejected from day 1 and 2, respectively. No epochs were rejected from the able-bodied participants. The same number of idle epochs was extracted to match the number of movements for each participant to have the same number of movement intention and idle activity epochs.

### 2.5. Feature Extraction

Three types of features were extracted from the movement intention epochs and idle activity epochs: temporal, spectral and template matching features, which are similar to previous works [33,36]. The features were extracted from the five EEG channels. The temporal features were the mean amplitude in 0.5 s non-overlapping windows, i.e., four features were extracted, and the fifth temporal feature was the difference in mean amplitude between the first and second half of the epoch. Twenty-three spectral features were calculated per channel. The power spectral density was calculated using a Hamming window with 0.5 s overlap between 8 and 30 Hz in 1 Hz bins. Lastly, a template matching feature was calculated per channel. The template was calculated as the mean across all movement intention epochs in the training data set. The cross correlation between the template and each epoch was calculated, and the value with zero time lag was used as a feature.

### 2.6. Classification Scenarios

The features were classified using a random forest classifier with 128 trees [47]. Four different classification scenarios were investigated where the classifier was calibrated and tested in different ways.

#### 2.6.1. Within-Session

The within-session classification scenario was investigated where data from the same participant in the same session was used for training and testing, using leave-one-sample-out cross-validation. This analysis was performed for each day for the able-bodied participants performing wrist extensions, and for each day for the CP participants performing wrist extensions and ankle dorsiflexions.

#### 2.6.2. Between-Session

The between-session classification scenario was investigated where data from the same participant but from different sessions were used for training and testing. For the able-bodied participants, data from two of the sessions was used for training and data from the last session was used for testing; the different combination of sessions was evaluated. Similarly for the CP participants, both combinations of constructing training and testing data were evaluated, i.e., training on data from day one and testing on data from day two and vice versa. CP participant 5 was not included in this analysis since only one recording session was available.

#### 2.6.3. Across-Participant

In the across-participant classification scenario, data from all sessions from all participants except the sessions from one participant was used for training the classifier in a leave-one-participant out cross-validation scheme. The classifier was tested on the data from the left-out participant’s sessions, of which there were two or three depending on the health condition (able-bodied or CP participant). The data from the able-bodied participants and CP participants was not mixed.

#### 2.6.4. Across-Condition

In the across-condition classification scenario, data from all sessions from all able-bodied participants were used for training the classifier. The classifier was tested on the data from the CP participants, comprising both data associated with wrist extension and ankle dorsiflexions.

### 2.7. Feature Analysis

To estimate the importance of the three feature types, i.e., temporal, spectral, and template matching, for the classification of movement intention and idle activity epochs, each feature type was used individually in the within-session classification scenario described above, where leave-one-sample-out cross-validation was used.

### 2.8. Statistical Analysis

Two 1-way repeated measures analysis of variance (ANOVA) tests were performed with classification scenario as the factor for CP participants (4 levels: within-session, between-session, across-participant, and across-condition) and able-bodied participants (3 levels: within-session, between-session, and across-participant). The data were pooled across days. Moreover, two 1-way repeated measures ANOVA tests were performed for CP and able-bodied participants with feature type as the factor (3 levels: temporal, spectral, and template matching). The data were pooled across days. When the assumption of sphericity was violated, a Greenhouse–Geisser correction was applied. Significant tests were followed up with post hoc analysis using Bonferroni correction to account for multiple comparisons. Statistical significance was assumed when *p* < 0.05 in all tests.

## 3. Results

The grand averages across participants are summarized in Figure 2. The grand averages of the movement intention epochs show the expected increase in negativity towards the movement onset. The amplitude of the most negative part of the movement intention was around 5 µV for the wrist extensions and close to 10 µV for the ankle dorsiflexions for the CP participants with slightly larger inter-participant variability on day 1 compared to day 2. Similarly, amplitudes reaching 10 µV were observed for the able-bodied participants performing wrist extensions across the three days. The amplitudes of the movement intentions were similar for Cz and C3.

The results of the classification analysis are summarized in Figure 3 and Table 2. The highest classification accuracies were obtained in the within-session classification scenario with classification accuracies around 75% and 85% for CP and able-bodied participants, respectively. In the between-session classification scenario, the performance dropped by approximately 15 percentage points for the CP participants to ~60% while the performance of the able-bodied participants was maintained around 80–85%. The performance decreased further with a few percentage points in the across-participant classification scenario for both CP and able-bodied participants. Lastly, in the across-condition classification scenario, where the data from able-bodied participants was used to calibrate the classifier, the performance was similar to the between-session and across-participant classification scenarios with accuracies around 60% for the CP participants. The statistical analysis revealed a significant effect of classification scenario for CP participants (F_(3,45)_ = 13.4; *p* < 0.001; η^2^ = 0.47) and able-bodied participants (F_(1.6,70.7)_ = 13.5; *p* < 0.001; η^2^ = 0.23). The post hoc analysis showed that significantly higher classification accuracies were obtained for the within-session classification scenario compared to the others for the CP participants. For the able-bodied participants, each classification scenario was significantly different from the others, with the highest and lowest classification accuracies obtained for within-session and across-participant classification scenarios, respectively.

Generally, an equal number of true positive rates and true negative rates were obtained for the CP- and able-bodied participants except in the across-participant and across-condition calibration scenarios, where the true negative rates were higher than the true positive rates for the CP participants. In these scenarios, the false negative rates were also higher than the false positive rates.

For the feature analysis (summarized in Figure 4 and Table 3), similar classification accuracies were obtained using the three feature types for the CP participants. In the statistical analysis, there was no significant effect of feature type for the CP participants (F_(1.5,24.9)_ = 0.3; *p* = 0.70; η^2^ = 0.02). For the able-bodied participants, the statistical analysis revealed a significant effect of feature type (F_(1.1,48.7)_ = 80.1; *p* < 0.001; η^2^ = 0.65). Significantly higher classification accuracies were obtained using temporal and template matching features compared to spectral features.

## 4. Discussion

The results of the classification analysis revealed that it is possible to discriminate between movement intentions and idle brain activity from single-trial EEG in people with CP, showing that it is technically feasible to perform BCI training for inducing neural plasticity in individuals with CP of the upper and lower extremities. The highest accuracies for the CP participants were obtained for the within-session classification scenario, with accuracies reaching 75%. The performance dropped for the other classification scenarios. The performance associated with the within-session classification scenario in this study is in the range of what has been reported in similar studies with CP participants, where accuracies around 70–80% were reported [11,37,38]. The performance of the CP participants was lower compared to the able-bodied participants. This is consistent with findings where BCI performance has been compared between stroke patients and able-bodied participants [25,29]. The differences in classification accuracies between the CP participants and able-bodied participants in the current study can be explained by the smaller amplitudes of the movement intentions in CP participant compared to the able-bodied participants; consequently, the amplitude differences between movement intentions and idle brain activity are smaller for the CP participants (see Figure 2). Also, the differences in classification accuracies may be attributed to a difference in the strength of the ERD, which has been reported to be weaker in people with CP [48], although comparable ERD patterns between children with CP and typically developing children have been reported [49]. The better performance of the within-session classification scenarios compared to the other classification scenarios is consistent with evidence from movement intention detection in people with stroke, spinal cord injury, Parkinson’s disease, and multiple sclerosis [33,36,45], which suggests that training data from the same session is needed, at least for CP participants. The findings from able-bodied participants showed that good performance can be maintained in the between-session classification scenario which is likely due to a similar morphology of the MRCPs across days. A between-session calibration scheme could potentially be improved for CP patients once they start training and produce more consistent MRCPs and stronger ERD patterns [10]. Moreover, a generalized classifier could potentially be used to avoid spending time on initial calibration using transfer learning [13]. The performance would initially be lower, but it could be adapted continuously during the training session, and the performance would improve [44,45,50]. The adaptation could, e.g., be performed in cue-based BCI systems when it is known when the user attempts to activate the BCI, or by detecting error-related potentials which could be used for labeling the data used for recalibrating the classifier [51]. Also, the data could be labeled by a separate classifier that used the entire waveform of the MRCP [24] and post-movement beta event-related synchronization [22,52], which would make the detection more robust.

The classification accuracies for the different classification scenarios were significantly higher than chance level calculated with a significance level of 5% for the able-bodied participants [53]. The within-session classification accuracies for the classification scenario for the CP participants were significantly higher than chance level, but the classification accuracies associated with the other classification scenarios were not. As outlined, the performance could be improved by recalibrating the classifier with new training data or potentially by identifying features and classifiers that are robust towards the intra- and intersubject variability. The findings in the current study suggest that the different feature types contribute, along with complementary discriminative information, to the classification, which is also supported by previous work where feature importance was estimated [37]. Spatial filtering can potentially improve the signal-to-noise ratio and hence improve the detection performance [29]; alternatively, other classifiers such as simple linear classifiers with good generalization properties or deep learning techniques could potentially improve this. It has been suggested in a recent study that the non-deep learning techniques outperform the deep learning techniques [54], but in order to use the deep learning techniques there needs to be enough training data to tune the hyperparameters.

The estimated detection performance in the current study is sufficient for inducing neural plasticity when combined with, e.g., electrical stimulation nerve stimulation. The lower limit of BCI performance required for using the BCI for inducing plasticity is not known, but it has been shown that plasticity can be induced with movement intention detection performance in the range of 65–85% [14,15,16,17]. It has been suggested that the BCI performance is positively correlated with the induction of plasticity, so there is an incentive to improve the BCI performance, although mixed results have been reported [15,17]. Despite modest movement detection performance, it can be used for inducing plasticity and hence promote motor learning. Improving BCI performance, however, could be an important aspect since the user may become frustrated and disengaged if the performance is poor, and it will increase the risk of the user discontinuing the training. This issue can be mitigated by gamifying BCI training where poor performance can be concealed and alleviated using performance accommodation mechanisms and other game mechanics to modulate the user’s perceived control and frustration [55].

The findings in the current study are in agreement with previous work within movement intention detection with perspectives for BCI training for inducing plasticity, but there are some limitations of this work. The detection performance is estimated by classifying between idle brain activity epochs and movement intention epochs that are extracted with a priori knowledge of when the movement occurs. This information is not known in an online BCI, and it is expected that the reported offline analysis performance will be better than that obtained when using an online BCI system. Also, the classification will occur continuously when EEG is recorded, which requires that decision rules or thresholds must be set to obtain a desired trade-off between true positive rate and false positive rate [25]. For a clinical outcome, it is hypothesized that it is more important to maximize the true positive rate and accept a higher false positive rate. However, too many false positive detections could also lead to a lower agency for the user. This issue can potentially be mitigated to some degree by only accepting BCI input in pre-defined time windows [6]. Another limitation of this study was that no EMG was recorded from the able-bodied participants, which can lead to a slightly different synchronization between movement onsets and the EEG. This makes it more difficult to compare the able-bodied participants with the CP participants. However, based on Figure 2, the able-bodied participants were able to accurately time their movement onsets to the visual cue. The grand average MRCPs in Figure 2 look different for CP and able-bodied participants, but it should be noted that the MRCP generally shows a considerable inter-participant variability. It is hypothesized that the grand average MRCP for the CP participants (*n* = 5) would resemble that of the able-bodied participants (*n* = 15) if the same number of CP participants would have been included in the study. Including more participants would also increase the generalization of the results. Individuals with CP have a wide spectrum of motor impairments, and more participants would be needed to investigate if the level of motor impairment correlates with the BCI performance. Moreover, the data set from the CP population only included male participants, which limits the generalization of the results to female individuals with CP. However, based on previous work, it has been reported that gender does not affect BCI performance when controlling the BCI with motor imagery [56].

The approach in the current study has potential for being adopted in clinical practice since the feature extraction and calibration of the classifier is simple and takes very little time. It means that the helpers setting up the BCI do not need to spend much time on system calibration, which is important if the users have a limited attention span. Moreover, with a setup with a low number of electrodes, it could reduce the need for washing hair after the training, which is desirable if it is being used daily. Despite the potential for adopting BCI in clinical practice, it should be tested whether it can be used for inducing neural plasticity, and how important the delay is between the movement intention and inflow of afference. If the delay is in the order of 200–300 ms, then it should be considered to use EMG to trigger the afferent feedback if the CP patient is able to produce that. In a study comparing EEG- and EMG-triggered electrical stimulation, it was shown that the two systems induced neural plasticity to the same degree in able-bodied participants [15].

In conclusion, movement intentions can be detected from single-trial EEG in people with CP, showing the technical feasibility of using BCI training for motor habilitation in people with CP. It is suggested that the BCI is calibrated immediately before each training session to obtain good BCI performance. In future studies, the findings should be validated with more patients using an online BCI that is combined with, e.g., electrical stimulation to investigate whether there is a clinically significant effect of BCI training in a CP population.

## Figures and Tables

**Figure 1 sensors-25-07347-f001:**
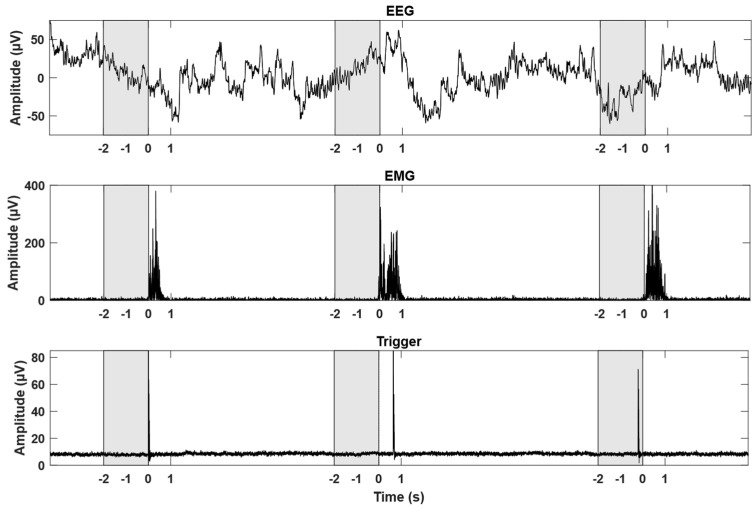
The top graph shows the filtered continuous EEG from Cz, while the filtered and rectified EMG is shown in the middle graph (two repetitions of a movement), and the digital trigger from the cuing program is shown in the bottom graph. The x-axis was adjusted according to the detected movement onsets from the EMG. The shaded gray area represents the movement intention epochs that were extracted from the continuous EEG.

**Figure 2 sensors-25-07347-f002:**
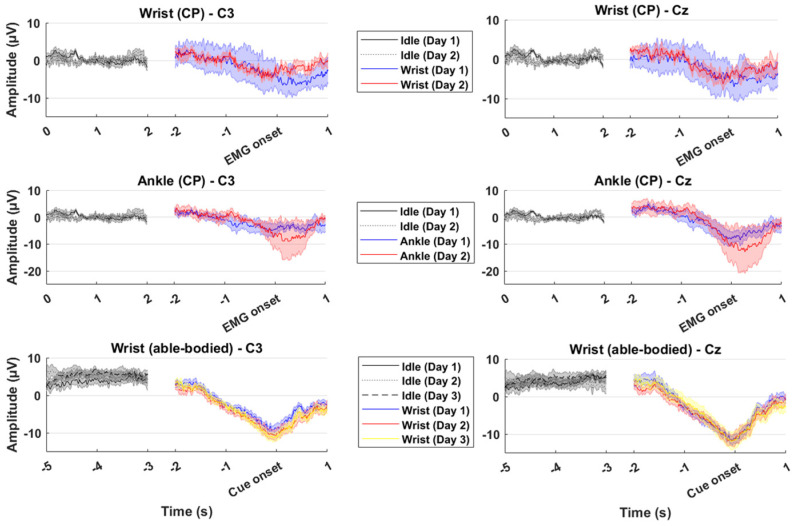
Grand average across participants of the idle activity epochs and movement intention epochs. The two top graphs are wrist extensions from the participants with cerebral palsy (CP) from channel C3 and Cz, respectively. The two middle graphs are ankle dorsiflexions from the participants with CP from channel C3 and Cz, respectively. The two bottom graphs are wrist extensions from the able-bodied participants from channel C3 and Cz, respectively. The shaded areas on the graphs show the standard error across participants. Note the difference in the y-axes from the ankle dorsiflexions and the wrist extensions.

**Figure 3 sensors-25-07347-f003:**
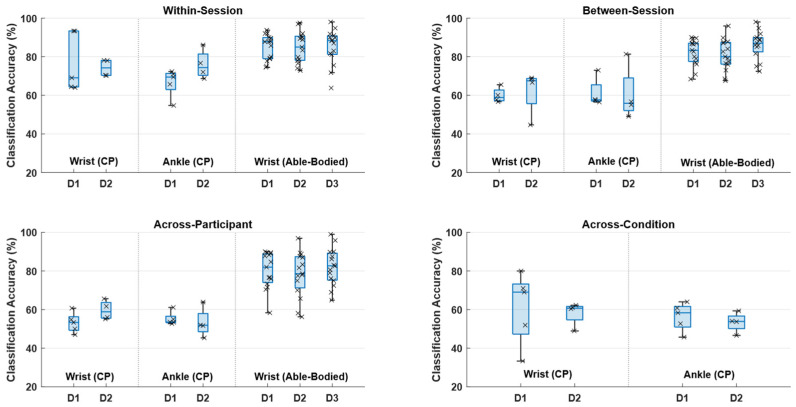
The figure shows the boxplots and individual classification accuracies for each participant for the four different classification scenarios. ‘D’ indicates the day. In the between-session graph (top right corner), D1 refers to the testing set, i.e., the classifier was trained on data from day 2 for the cerebral palsy (CP) participants or on data from day 2 and 3 for the able-bodied participants.

**Figure 4 sensors-25-07347-f004:**
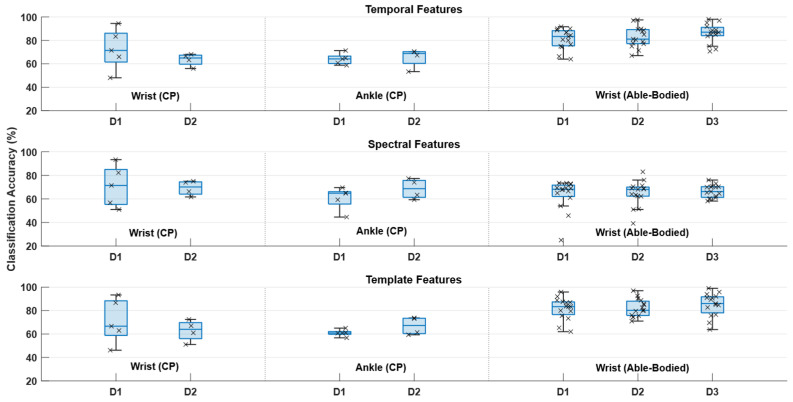
The figure shows the boxplots and individual classification accuracies for each participant for the within-session calibration scenario when using the individual feature types, i.e., temporal, spectral, and template matching features. ‘D’ indicates the day. CP: cerebral palsy.

**Table 1 sensors-25-07347-t001:** Overview of the characteristics of the participants with cerebral palsy (CP). To provide an indication of the participants’ ability to use their arms they were asked the following question: Can you wash your hair? With the following response options: Yes with 2 hands, Yes with 1 hand, or No. To provide an indication of the participants’ ability to use their legs, their ability to walk and whether they use walking aids is also indicated in the table.

Patient	Age	Gender	Diagnosis	Most Affected Side	Able to Wash Hair	Walking Ability
1	50	Male	CP (Diplegia)	Right	Yes, 2 hands	Yes, with cane
2	40	Male	CP (Tetraparesis)	Right	No	No, sits in electrical wheelchair
3	34	Male	CP (Tetraparesis)	Right	No	No, sits in electrical wheelchair
4	19	Male	CP	Right	Yes, 2 hands	Yes, with walker
5	22	Male	CP (Hemiplegia)	Right	Yes, 2 hands	Yes

**Table 2 sensors-25-07347-t002:** The confusion matrices of the different calibration scenarios are presented across participants. The true labels are displayed on the rows, and the predicted labels are displayed on the columns. Row 1 and column 1: true positive rate. Row 1 and column 2: false negative rate. Row 2 and column 1: false positive rate. Row 2 and column 2: true negative rate. All values are in percentage.

Within-Session	Day 1	Day 2	Day 3
Wrist (CP)	75 25	76 24	
	27 73	27 73	
Ankle (CP)	64 36	75 25	
	32 68	25 75	
Wrist (Able-Bodied)	85 15	85 15	87 13
	16 84	16 84	15 85
**Between-Session**	**Day 1**	**Day 2**	**Day 3**
Wrist (CP)	53 47	63 37	
	33 67	38 62	
Ankle (CP)	53 47	65 35	
	30 70	42 58	
Wrist (Able-Bodied)	81 19	81 19	88 12
	16 84	19 81	15 85
**Across-Participant**	**Day 1**	**Day 2**	**Day 3**
Wrist (CP)	44 56	53 47	
	38 62	34 66	
Ankle (CP)	48 52	40 60	
	38 62	35 65	
Wrist (Able-Bodied)	78 22	79 21	**81 19**
	16 84	22 78	16 84
**Across-Condition**	**Day 1**	**Day 2**	
Wrist (CP)	50 50	50 50	
	33 67	34 66	
Ankle (CP)	49 51	40 60	
	37 63	32 68	

**Table 3 sensors-25-07347-t003:** The confusion matrices across participants of the within-session classification scenario using temporal, spectral, and template matching features as input for the classifier. The true labels are displayed on the rows and the predicted labels are displayed on the columns. Row 1 and column 1: true positive rate. Row 1 and column 2: false negative rate. Row 2 and column 1: false positive rate. Row 2 and column 2: true negative rate. All values are in percentage.

Temporal	Day 1	Day 2	Day 3
Wrist (CP)	69 31	63 37	
	29 71	36 64	
Ankle (CP)	64 36	61 39	
	37 63	32 68	
Wrist (Able-Bodied)	82 18	83 17	89 11
	21 79	19 81	15 85
**Spectral**	**Day 1**	**Day 2**	**Day 3**
Wrist (CP)	66 34	72 28	
	33 67	34 66	
Ankle (CP)	60 40	69 31	
	40 60	32 68	
Wrist (Able-Bodied)	66 34	67 33	69 31
	34 66	36 64	35 65
**Template**	**Day 1**	**Day 2**	**Day 3**
Wrist (CP)	68 32	64 36	
	32 68	39 61	
Ankle (CP)	59 41	67 33	
	38 62	35 65	
Wrist (Able-Bodied)	81 19	83 17	87 13
	20 80	19 81	15 85

## Data Availability

The raw data supporting the conclusions of this article will be made available by the authors upon reasonable request.

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
