# Peer review of "Comparison of Classifier Calibration Schemes for Movement Intention Detection in Individuals with Cerebral Palsy for Inducing Plasticity with Brain–Computer Interfaces"

_sensors, 2025, doi:10.3390/s25237347_

Round 1
Reviewer 1 Report
Comments and Suggestions for Authors
A brief summary
The paper investigates the effectiveness of decoding movement intentions from single-trial EEG data in individuals affected by cerebral palsy (CP). Experiments involved 5 participants with CP and 15 able-bodied control participants. During the experiments, the authors evaluated the accuracy of movement intention detection across four different calibration scenarios. The authors showed that BCI training is technically feasible for motor habilitation in people with CP. The manuscript is clear and presented in a well-structured manner.
General concept comments
Article
The manuscript is scientifically sound, and the experimental design is appropriate to test the hypothesis. The study concluded with insights into possible methods of employing BCI training to promote neural plasticity.
Review
The cited references are relevant, and 13 out of 59 references have been published within the last 5 years. The manuscript contains 19 self-citations.
The authors demonstrated that the classification accuracies across the 4 scenarios, as well as the main conclusions regarding BCI calibration options, are consistent with findings from similar studies cited in the paper. On the one hand, this indicates the proper application of the random forest classifier and suggests accurate feature extraction from single-trial EEG data. However, on the other hand, the novelty of the presented research remains unclear. The experimental design involving the four proposed scenarios does not seem to introduce new methodological approaches. The literature also widely discusses different classifier calibration methods across various BCI paradigms, including those based on machine learning and deep learning. As a result, evaluating the relative impact or novelty of the proposed study in relation to alternative research approaches is challenging.
If the novelty of the work lies solely in the use of BCI training to induce neural plasticity specifically in individuals with cerebral palsy, this should be explicitly stated in the Abstract, Introduction, and Discussion. In this case, it is essential to explicitly demonstrate how the obtained results differ from those of similar studies involving individuals with cerebral palsy, as referenced by the authors. This approach may be useful and of interest only to a narrow research community.
In other cases, the validation of the proposed investigation would benefit from a more comprehensive statistical assessment and analysis. The authors should way better describe and emphasize the advantage of the investigation they proposed. For example, for the machine learning classification, temporal, spectral, and template-matching features were engineered. A further analysis to determine which types of features are most important for individuals with CP and for able-bodied participants – particularly whether these feature types are common to both groups– would not only be of interest but also serve to complement the research findings.
Specific comments
- To evaluate false positive detections (page 9) more comprehensively, it is advisable to also employ the F-measure.
- It is advisable to at least briefly justify the selection of the five EEG channels over the central area of the international 10-10 system.
- In the opinion of the reviewer, the title of the article does not adequately reflect the primary aim of the work, which involves "comparing different calibration scenarios to investigate the decoding of movement intentions from single-trial EEG in individuals with cerebral palsy and able-bodied participants, with an emphasis on using these findings to induce neural plasticity".

Reviewer 2 Report
Comments and Suggestions for Authors
This paper is valuable in verifying the feasibility of using brain-computer interfaces (BCI) for detecting the movement intentions of patients with cerebral palsy. There are some comments:
- Only 5 patients with cerebral palsy were included, all of whom were male. The sample size was far smaller than the scale required by clinical studies in general. Moreover, there was no data on female patients, which prevented the coverage of the gender diversity of cerebral palsy patients. As a result, the conclusion could not be generalized to a broader population of cerebral palsy patients.
- Among the 5 patients with cerebral palsy, all were affected by the right side of the body, which does not represent the diversity of the pathological conditions of cerebral palsy patients. This weakens the generalizability of the research conclusion.
- Although time-domain, frequency-domain and template-matching features were extracted, how to fusion these features? In addition, no analysis was conducted on which features contributed the most to the classification.
Reviewer 3 Report
Comments and Suggestions for Authors
The paper concerns the developing field of using BCI systems in motor rehabilitation of people suffering from cerebral palsy. I have the following questions and comments on the presented study.
The Introduction may be improved by referencing clinical studies of using BCIs to improve motor functions in patients with cerebral palsy.
It may be better to present the able-bodied participants' quartiles rather than mean\std to make comparison between two groups of participants easier.
Typical resulting sizes of training and testing sets for each scenario should be presented. Also, please report the percentages of the rejected epochs.
No statistical analysis was performed to reveal which of the observed differences in classification accuracy were significant. One can, of course, guess that the most prominent differences were significant (e.g. 60% vs 85%), however the significance is not so evident for within-session or across-session scenarios.
It is not clear what rehabilitation scenario is proposed in the paper. What epochs are to be classified during the clinical procedure? It seems that two different phenomena were confused. The MRCP, observed prior to movement onset in the CP group, and something like contingent negative variation ("something like" because there apparently were no warning stimuli in the experiment) which might reflect action preparation and anticipation in able-bodied subjects prior to the cue onset, likely due to the fixed time interval between the cues. The confusion may result from unclear description of what the trigger is. From the text it seems that trigger was synchronized with visual go signal, however the figure 1 shows significant delay of the trigger from the movement onset. If the trigger indeed was synchronized with the go signal, then the signal trends observed in healthy group and reported in figure 2 are not MRCPs and therefore should not be compared to the epochs obtained in the CP group prior to movement onset. Therefore I suggest clarifying the proposed rehabilitation procedure design (i.e. that signals are classified before the cure and if the negativity is detected then the movement is assisted immediately after the signal or that the signals are classified according to the movement onset). I also suggest to use the same onset for all groups, either the cue or the movement onset. Since no movement onset data are available for the healthy subjects, it may be enough to use some average delay. Nevertheless, it should be explained in more detail why the movement onset was not measured in healthy group, I think it is a critical limitation of the study.
In case it is the MRCP which is to be detected by the proposed BCI, inevitably requiring measuring or estimating the movement onset during the training phase at least, it should be discussed what advantages the EEG system has over the EMG interface which just detects the movement onset and sends the feedback with little delay or robot-assisted rehabilitation (yes, the importance of little delay is discussed in the introduction, however in the majority of post-stroke BCI RCTs the systems with inherent delays were still used). In case the cue anticipation and possible related movement preparation is to be detected, the introduction should be re-written and the CP data should be processed accordingly.
Round 2
Reviewer 1 Report
Comments and Suggestions for Authors
The authors implemented substantial revisions, resulting in a significant improvement of the manuscript. In the revised version of the manuscript, the theoretical descriptions of the proofs are precise and rigorously defined. The authors clearly acknowledged the limitations of the methodology and articulated the primary objective of the work. They included additional statistical analyses, which confirmed the feasibility and effectiveness of BCI training in CP patients. The obtained results may prove beneficial to the research community.

Reviewer 3 Report
Comments and Suggestions for Authors
My comments were addressed and ny questions were answered.